# Effects of Cannabidiol and Beta-Caryophyllene Alone or in Combination in a Mouse Model of Permanent Ischemia

**DOI:** 10.3390/ijms22062866

**Published:** 2021-03-11

**Authors:** Cody G. Yokubaitis, Hassan N. Jessani, Hongbo Li, Allison K. Amodea, Sara Jane Ward

**Affiliations:** Center for Substance Abuse Research, Department of Pharmacology, Lewis Katz School of Medicine, Temple University, Philadelphia, PA 19140, USA; yok@temple.edu (C.G.Y.); hassan.jessani@temple.edu (H.N.J.); hongbo.li@temple.edu (H.L.); allisonamodea@gmail.com (A.K.A.)

**Keywords:** stroke, cannabidiol, beta-caryophyllene, microglia, photothrombosis, cannabinoid CB2 receptor

## Abstract

Current treatments for stroke, which account for 6.5 million global deaths annually, remain insufficient for treatment of disability and mortality. One targetable hallmark of stroke is the inflammatory response following infarct, which leads to significant damage post-infarct. Cannabinoids and their endogenous targets within the CNS have emerged as potential treatments for neuroinflammatory indications. We and others have previously shown that synthetic agonists of the cannabinoid CB2 receptor reduce infarct size and microglial activation in rodent models of stroke. The non-cannabinoid receptor mediated effects of the phytocannabinoid cannabidiol (CBD) have also shown effectiveness in these models. The present aim was to determine the single and combined effects of the cannabis-derived sesquiterpene and putative CB2 receptor agonist β-caryophyllene (BCP) and CBD on permanent ischemia without reperfusion using a mouse model of photothrombosis. Because BCP and CBD likely work through different sites of action but share common mechanisms of action, we sought to determine whether combinations of BCP and CBD were more potent than either compound alone. Therefore we determined the effect of BCP (3–30 mg/kg IP) and CBD (3–30 mg/kg IP), given alone or in combination (30:3, 30:10, and 30:30 BCP:CBD), on infarct size, microglial activation, and motor performance.

## 1. Introduction

Strokes account for 6.5 million global deaths annually. In the United States, on average someone suffers a stroke every 40 s, and every four minutes someone dies of one [1]. Patients that survive the initial episode experience a 40% increase in death, dependence, or institutionalization in the three months following [2]. The most common cause of stroke is an ischemic episode caused by a thrombus or embolism. Current treatments involve thrombolytic agents, such as tissue plasminogen activator (tPA) which provides a 30% improvement in disability outcomes when given within 3–4.5 h following an ischemic episode [3]. Importantly, another targetable hallmark of stroke is activation of the inflammatory response following infarct which leads to further significant damage [4,5,6]. For example, microglia, as the main resident immune cells, may display a beneficial effect in ameliorating secondary injury following stroke [7,8], with current research focusing on shifting polarization from pro-inflammatory phenotype to anti-inflammatory phenotype as a novel therapeutic target for post-stroke neurological recovery [9,10]. Prior experiments with mouse models from our laboratory and others have shown that targeting the immune response with cannabinoids can reduce infarct volume and improve behavioral outcomes [11,12,13,14].

The canonical cannabinoid CB1 and CB2 receptors are currently under investigation for a range of neuroinflammatory disorders [15]. The CB1 receptor is ubiquitously located throughout the nervous system, and activation of the CB1 receptor is associated with psychoactive effects that have historically limited the clinical application of this cannabinoid target [16]. By contrast, the CB2 receptor is localized predominantly to non-neuronal tissue, including immunomodulatory cells like microglia, astrocytes, and endothelial cells. Activation of CB2 receptors produces positive immunomodulatory role in broad spectrum of rodent models including multiple sclerosis [17], spinal cord injury [18], and rheumatoid arthritis [19]. As mentioned above, previous work from our laboratory have also shown that selective CB2 receptor agonists reduce infarct size, improve cognitive and motor outcomes, and attenuate neuroinflammation in mouse models of both transient middle cerebral artery occlusion (MCAO) [11] and permanent ischemic photoinjury [14].

Beta-caryophyllene (BCP) is a sesquiterpene found in *Cannabis sativa* and several other plant species such as black pepper, clove, rosemary, and hops. The pharmacological properties of BCP have been studied for decades and include antioxidant, anti-inflammatory, antimicrobial, cardioprotective, and neuroprotective effects (for review see [20]). Gertsch et al. in 2008 [21] reported selective CB2 receptor agonist activity of BCP in nanomolar concentrations. Indeed, several of BCP’s anti-inflammatory and neuroprotective effects in animal models have been shown to be reversed by administration of selective CB2 receptor antagonists, suggesting this mechanism is at least in part responsible for BCP’s immunomodulatory effects [22,23]. Using a mouse model of MCAO, Tian and colleagues demonstrated that BCP administration reduced infarct size and microglial activation and decreased the expression of toll-like receptor 4 (TLR4), interleukin-1β (IL-1β), and tumor necrosis factor-α (TNF-α) levels [24,25].

Cannabidiol (CBD) is a non-psychoactive component of *Cannabis sativa* that produces a wide range of pharmacological effects independent of cannabinoid receptors [26]. Basic and/or clinical studies have shown that CBD can produce antioxidant, anti-inflammatory, immunomodulatory, antiarthritic, anticonvulsant, neuroprotective, procognitive, anti-anxiety, antipsychotic and anti-proliferative effects, among others [27,28]. Not surprisingly then, CBD has a complex pharmacodynamic profile. For example, CBD acts as an agonist at the following receptors: transient receptor potential ankyrin subfamily member 1 (TRPA1) and vanilloid subfamily members 1–4 (TRPV1–4), peroxisome proliferator-activated receptor γ (PPARγ), orphan G-protein coupled receptors GPR55 and GPR18, and serotonin 5-HT_1A_ and 5-HT_2A_ receptors (partial agonist). Moreover, CBD is a positive allosteric modulator of α1-, α1β- and α3-glycine receptors (α1-, α1β- and α3-GlyR), μ- and δ-opioid receptors (μ- and δ-OR) and γ-aminobutyric acid receptor type A (GABA_A_) [29]. Regarding stroke, Hayakawa et al. have shown in a series of studies using a mouse model of MCAO that CBD administration reduced infarct size, possibly through 5-HT1A receptor and other mechanisms [30,31,32,33].

The aim of the present study was to determine the effect of BCP or CBD on permanent ischemia without reperfusion using a mouse model of photothrombosis, with a focus on potential interactive effects of these two compounds. Because BCP and CBD likely work through different sites of action but share common mechanisms of action (e.g., anti-inflammatory, anti-oxidant), we sought to determine whether combinations of BCP and CBD were more potent or produced different effects than either compound alone. Increased efficacy and potency of drug combinations leading to lower required doses of one or both agents is associated with an improved therapeutic index. While no adverse effects of BCP in humans have been identified, CBD at higher doses has been associated mild adverse effects such as GI upset and somnolence, to more rare but severe adverse effects such as hepatotoxicity. Therefore we determined the effect of BCP (3–30 mg/kg IP) and CBD (3–30 mg/kg IP), given alone or in combination (30:3, 30:10, and 30:30 BCP:CBD), on infarct size, microglial activation, and motor performance. A peri-injury dosing regime of 1 h pre- and 24 h post-infarct was employed to replicate previous findings from our group with other cannabinoid compounds.

## 2. Results

### 2.1. Descriptive and Statistical Findings

#### 2.1.1. Infarct Size

I.Treatment with BCP, CBD, and their combination led to a significant decrease in infarct size.II.Following administration of BCP, one-way ANOVA test revealed a significant effect of BCP (F (3, 28) = 5.237, *p* = 0.005), with Dunnett’s multiple comparison showing a significant effect at the 30 mg/kg dose (Figure 1A).III.Following administration of CBD, one-way ANOVA revealed a significant effect of CBD (F (3, 27) = 3.974, *p* = 0.018), with Dunnett’s multiple comparison test showing a significant effect at the 30 mg/kg dose.IV.Following administration of BCP:CBD combinations, one-way ANOVA revealed a significant effect of BCP (F (3, 27) = 7.182, *p* = 0.001), with Dunnett’s multiple comparison test showing a significant effect at the 30:3 and 30:30 BCP:CBD mg/kg doses (Figure 1C).V.Figure 1D shows the location and shape of the infarct in coronal section and its attenuation by BCP, CBD, and the combination in representative mice. The center of the lesion is in parietal cortex area 1 (primary somatosensory cortex) at approximately –2.0 mm from Bregma, and the extent of the infarct can be observed in the coronal slices from approximately 0.20 mm rostral to Bregma to approximately –2.80 mm caudal to Bregma. The infarct spreads to the border of the corpus callosum but is not observed in deeper tissue such as the CA1-3 fields of the hippocampus as measured with the TTC staining. Interestingly, BCP treatment led to more shallow but broader damage, CBD treatment led to a narrower, but deep, infarct, and the combinations combined these two characteristics with a more narrow and shallow range of damage.

#### 2.1.2. Microglia Analysis

Total fluorescence: Increased immunolabeling for Iba-1 was observed in the stroke animals compared to sham control (Figure 2A–C, Student *t*-test sham versus vehicle, *p* < 0.05). The Iba1 antibody is commonly used as a marker of microglia activation in staining and immunohistochemistry, given that ionized calcium binding adaptor molecule 1 (Iba1) is a microglia/macrophage-specific calcium-binding protein with actin-bundling activity that participates in membrane ruffling and phagocytosis in activated microglia.
(1)Treatment with BCP or CBD alone did not significantly alter total fluorescence compared to the vehicle-treated stroked mice. One-way ANOVAs compared treatments in stroked animals, sham animals were excluded in the analysis. One way ANOVA for BCP was (F (3, 33) < 1.0), and one-way ANOVA for CBD was (F (3, 30) < 1.0) (Figure 2A,B).(2)However, treatment with the BCP:CBD combination led to a significant reduction in total immunofluorescence compared to vehicle-treated stroked mice, (F (3, 32) = 4.483, *p* = 0.01), with Dunnett’s multiple comparison test showing a significant effect at the 30:3 and 30:30 BCP:CBD mg/kg doses (Figure 2C).Microglial cell count. Increased Iba-1 positive cell number was observed in the stroke animals compared to sham control. (Figure 2D–F, Student *t*-test sham versus vehicle, *p* < 0.05).
(1)Treatment with BCP or CBD alone did not significantly alter total microglial cell number compared to the vehicle-treated stroked mice. One way ANOVA for BCP was (F (3, 31) < 1.0), and one way ANOVA for CBD was (F (3, 30) < 1.0) (Figure 2D,E).(2)Treatment with the BCP:CBD combination also did not alter total microglial cell number compared to vehicle-treated stroked mice (F (3, 30) = 1.042, *p* = 0.388) (Figure 2F).Microglial cell body size. Increased cell body size was observed in the stroke animals compared to sham controls. (Figure 2G–I, Student *t*-test sham versus vehicle, *p* < 0.05).
(1)Treatment with BCP produced a significant increase in average cell body size as compared with vehicle-treated stroked mice (F (3, 32) = 3.593, *p* = 0.0241) (Figure 2G), with Dunnett’s multiple comparison test showing no significant effect at any dose.(2)Treatment with CBD produced a significant increase in average cell body size as compared with vehicle-treated stroked mice (F (3, 32) = 2.728, *p* = 0.05) (Figure 2H), with Dunnett’s multiple comparison test showing a significant effect at the 3.0 and 30 mg/kg doses.(3)However, treatment with the BCP:CBD combination led to a significant reduction in average cell body size as compared with vehicle-treated stroked mice, (F (3, 32) = 6.319, *p* = 0.002), with Dunnett’s multiple comparison test showing a significant effect at the 30:30 BCP:CBD mg/kg doses (Figure 2I).

#### 2.1.3. Grip Strength Behavior

Decreased grip strength was observed in the stroke animals compared to sham controls.

(1)Treatment with BCP produced a significant improvement in grip strength as compared with stroke (F (3, 28) = 9.926, *p* < 0.0001) (Figure 3A), with Dunnett’s multiple comparisons test showing significance at the 3.0 and 10 mg/kg doses.(2)Treatment with CBD produced a significant improvement in grip strength as compared with stroke (F (3, 28) = 13.28, *p* < 0.0001) (Figure 3B), with Dunnett’s multiple comparisons test showing significance at the 30 mg/kg dose.(3)Treatment with the BCP:CBD combination produced a significant improvement in grip strength as compared with stroke (F (3, 28) = 8.535, *p* = *p* < 0.0001) (Figure 3C), with Dunnett’s multiple comparisons test showing significance at the 3.0, 10, and 30 mg/kg doses.

### 2.2. Figures, Tables and Schemes

**Figure 1 ijms-22-02866-f001:**
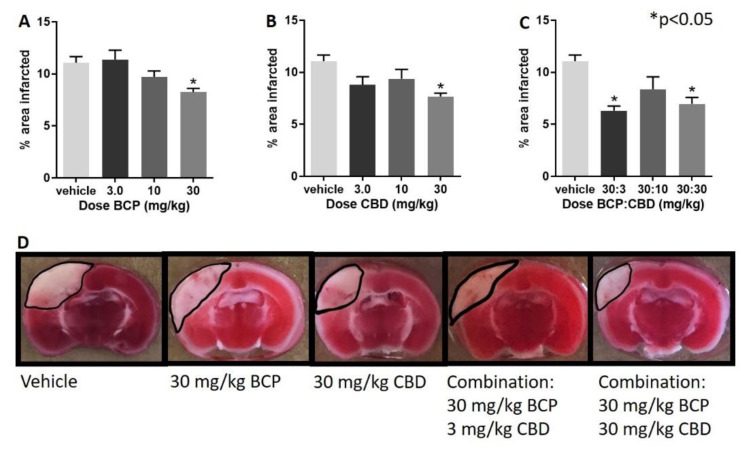
Beta-caryophyllene (BCP), cannabidiol (CBD) and BCP + CBD combinations significantly reduced infarct size in a mouse model of permanent ischemia (photoinjury). X axis: treatment (all mice exposed to photothrombosis and either vehicle or a dose of BCP, CBD, or their combination), Y axis, infarct area in mm2. (**A**) BCP treatment led to a significantly smaller infarct size as compared to vehicle treatment (* *p* = 0.005). (**B**) CBD treatment led to a significantly smaller infarct size as compared to vehicle treatment (* *p* = 0.018). (**C**) BCP + CBD treatment led to a significantly smaller infarct size as compared to vehicle treatment (* *p* = 0.001). Data are expressed as mean and SEM and were analyzed utilizing one-way ANOVA with Dunnett’s multiple comparison test. (**D**) The center of the lesion is in parietal cortex area 1 at approximately –2.0 mm from Bregma. The infarct size is measured with TTC staining. BCP treatment led to more shallow but broader damage, CBD treatment led to a narrower but deep, infarct, and the combinations combined these two characteristics with a more narrow and shallow range of damage.

**Figure 2 ijms-22-02866-f002:**
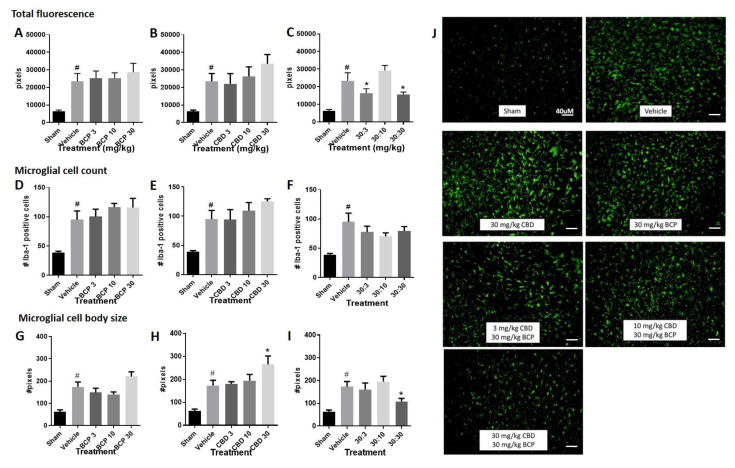
BCP or CBD single treatment increased Iba-1 immunofluorescence following photoinjury, while BCP + CBD combination treatment decreased Iba-1 immunofluorescence. (**A**–**C**) Quantification of Iba-1 immunofluorescence within the field of microscope images. (**D**–**F**) Quantification of microglial cell number within the field of microscope images. (**G**–**I**) Quantification of microglial cell body area (in arbitrary units determined using ImageJ) within the field of the microscope images. Data are expressed as mean and SEM. Statistical differences between sham (no stroke) and vehicle (stroke with vehicle treatment) were determined using Student t-test, ^#^
*p* < 0.05. Statistical differences across vehicle and cannabinoid-treated stroked mice were analyzed utilizing one-way ANOVA with Dunnett’s multiple comparison test to determine significance from vehicle treatment, * *p* < 0.05. (**J**) Representative immunohistochemical labeling for treatment groups are shown a 20× magnification.

**Figure 3 ijms-22-02866-f003:**
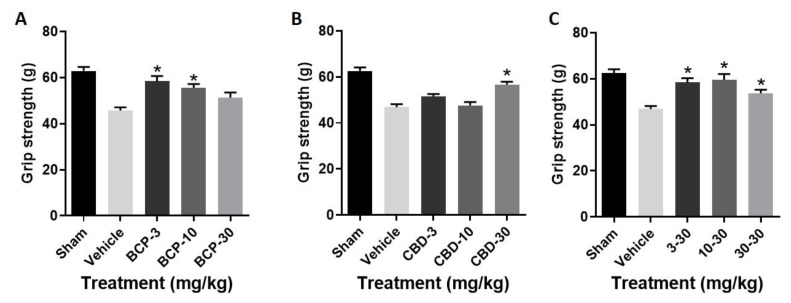
BCP, CBD and BCP + CBD combinations significantly improved grip strength in a mouse model of permanent ischemia (photoinjury). X axis: treatment, Y axis, grip strength in grams. (**A**) BCP treatment led to a significant improvement in grip strength as compared to vehicle treatment (* *p* = 0.0001). (**B**) CBD treatment led to a significant improvement in grip strength as compared to vehicle treatment (* *p* = 0.0001). (**C**) BCP + CBD treatment led to a significant improvement in grip strength as compared to vehicle treatment (* *p* = 0.0001). Data are expressed as mean and SEM and were analyzed utilizing one-way ANOVA with Dunnett’s multiple comparison test.

## 3. Discussion

In our study, we explored the effect of two “non-psychoactive” Cannabis constituents, CBD and BCP, alone and in combination, on photothrombotic stroke model. Both constituents have been researched in preclinical models of neuroprotection and inflammation and are gaining attention of clinicians for the therapeutic potential in humans. In our mouse model of permanent ischemia, photothrombosis was associated with significant cell death in the infarcted area, significant increases in Iba-1 staining (both total fluorescence and Iba-1 positive cell number) and increased cell body size, indicative a shift from the so-called “resting” ramified state to the “activated” amoeboid morphology. We determined that both constituents, when administered alone or as a combination, decreased infarct size, affected microglial activation, and improved motor performance. However, some of the combination effects showed a unique profile compared to single administration, suggesting that these compounds interact in ways which may improve their clinical utility and suggest mechanism(s) of action different than either agent alone.

As a plant derived, dietary CB2 receptor agonist, BCP’s ability to reduce infarct size and improve neurological performance is consistent with previous data from our laboratory with the synthetic CB2 receptor agonist O-1966 [14] and support our other work showing a protective role for CB2 receptor activation in rodent models of both permanent and transient ischemia [34,35,36]. In the present experiment, the 30 mg/kg dose of BCP significantly reduced infarct size. However, no BCP dose altered microglial number or immunofluorescence, and actually showed a trend toward increasing microglial activation. Moreover, only the two lower doses of BCP, but not the 30 mg/kg dose, improved grip strength behavior. These phenomena indicate that BCP treatment can produce protective effects from permanent ischemia through mechanisms not related to general microglial activation, and perhaps through a separate mechanism to reduce infarct size compared with motor protection. As mentioned previously, Tian et al. previously demonstrated using a mouse model of transient ischemia (MCAO), that BCP administration reduced infarct size, microglial activation, and a range of inflammatory mediators [24,25,37]. Their reported effect of BCP on microglial activity was at a larger dose of BCP (72 mg/kg) and was specific to driving microglial phenotype from the proinflammatory M1 to the anti-inflammatory M2 phenotype. Interestingly, Zarruk et al. [38] also reported that the CB2 receptor agonist JWH-133 decreased microglial activation, but found that both M1-type and M2-type microglial markers were attenuated. As we did not assess microglial gene expression changes in our present study, our lack of effect on Iba-1 staining could be due to not capturing changes in microglial phenotype. Alternately, protective effects of BCP at the doses tested here may be working through a microglia-independent mechanism. Taken together with the previous literature from our lab and others, these data strongly support a therapeutic role for CB2 receptor activation in the treatment of stroke.

As non- CB1 and CB2 receptor activating phytocannabinoid, CBD treatment also reduced infarct size and improved grip strength at the 30 mg/kg dose. CBD treatment also showed a similar pattern of increasing Iba-1 immunofluorescence we observed with BCP and did produce a significant increase in microglial cell body size at the 30 mg/kg dose, as is quite evident in the representative micrograph. Again this increase in cell body size is typically associated with a shift to a more “activated” microglilal phenotype. As mentioned in the introduction, CBD has also been shown to reduce infarct size in the transient MCAO model of ischemia [30,32,39]; however, microglial activation was not assessed in these studies. One report by Mori et al. [40] showed significant reduction in microglial reactivity by CBD in a mouse model of bilateral common carotid artery occlusion 21 days following injury, whereas our measurements were done only three days following induction of stroke. CBD has been reported to decrease microglial activation in other rodent models of nervous system injury, such as neuropathic pain [41] and newborn hypoxia [42]. In contrast, we found in a mouse model of spinal cord injury that while CBD decreased the development of spinal cord injury-associated neuropathic pain, T cell pathology, and the expression of several pro-inflammatory markers, it was not associated with a decrease in microglial activation [43]. Taken together these results again suggest that CBD’s neuroprotective or anti-inflammatory effects may be independent of effects on microglial activation or may be more nuanced toward altering microglial functionality that is not sensitive to characterization by use immunofluorescence with Iba-1 alone. It is also possible that CBD’s effects on microglial reactivity are time course dependent, as microglial responses to different injury types are heavily time course dependent.

At the outset of this study, we planned the single and combined doses based on the existing ischemia literature, which shows an inverted U-shaped dose response curve with CBD (effective doses 1.0 and 3.0 mg/kg IP, ineffective doses 0.1 and 10 mg/kg IP), and significant dose effects of BCP starting at 72 mg/kg but absent at lower doses. For combination groups, which we wanted to run concurrently with single dose animals, we chose the combinations of 30:3, 30:10, and 30:30 mg/kg BCP to CBD, presuming that we would be testing the hypothesis as to whether ineffective doses of BCP would alter the CBD dose response curve in the permanent ischemia model. However, as the 30 mg/kg dose of BCP alone was significant in reducing infarct size in this model, we were unable to test this specific hypothesis. That is to say, for each of our combination doses, at least one of the single agents was effective at that dose alone. Despite this drawback in experimental design, several very intriguing patterns of co-administration of BCP and CBD emerged. Firstly, only two of the dose combinations, 30:3 mg/kg and 30:30 mg/kg, significantly reduced infarct size, with the middle combination being ineffective, even though the dose of CBD in this combination was effective on its own. Similarly, this pattern was also observed in the microglial assessment, where the 30:3 and 30:30 mg/kg combinations significantly reduced total fluorescence, while the 30:10 mg/kg dose did not. Secondly, and perhaps more interesting, this effect (reducing total fluorescence and microglial cell body size with the combinations) is the opposite of what was observed regarding BCP and CBD increasing total microglial fluorescence and cell body size. Taken together these data highlight that in this instance, both increases and decreases in Iba-1 immunoreactivity are associated with a decrease in infarct size and improvement of grip strength. One conclusion to draw from this is that both directional effects of microglial activation represent protective phenotypic changes by these Cannabis constituents, perhaps mediated by different mechanisms. Future studies need to be designed to determine whether the combined effect of decreasing microglial activation rather than increasing it, as seen with the single compounds, affords any increased benefit to the injury that was not measured in the present study. We are currently following up on this research by analyzing a wide range of inflammation-related gene expression changes in single versus combined BCP and CBD treated mice following permanent ischemia, which can provide insight into more nuanced microglial phenotypic changes following stroke and cannabinoid treatment that were not captured in the present study. It will also be critical to assess these changes at different timepoints following ischemia, as microglial function can change immediately versus days to weeks post infarct. Lastly, the effectiveness of different treatment regimes, e.g., dosing animals at later time points post-infarct versus the peri-injury dosing regime used in the current study, will be important to determine.

## 4. Materials and Methods

### 4.1. Animals

This study was conducted in accordance with the guidelines approved by the Institutional Animal Care and Use Committee at Temple University (Animal Use Protocol #4498, approval date May 2018). Eighty male C57B/6 mice, 6–8 weeks old, and weighing 19–24 g were used for this study.

### 4.2. Induction of Stroke: Photothrombosis Model

Mice habituated to the animal facility for 5 days prior to onset of the surgical procedure. Vehicle, BCP, CBD, or their combination was administered IP 1 h before induction of stroke and 24 h post-stroke. Following the initial treatment or vehicle injection, mice were anesthetized by an IP injection of a mixture (1:1 by volume) of ketamine (100 mg/mL) and xylazine (20 mg/mL) at a dose of 1 mL/kg. Body temperature was monitored with a rectal thermometer and maintained between 36–38 °C with a warming pad during the surgical procedure and until adequate recovery. The scalp was excised over the skull and the periosteum removed. A marker was used to identify the sensorimotor cortex 2 mm posterior and 2 mm lateral to bregma. Rose Bengal (0.1 mL of 10 mg/mL) dissolved in saline was administered IP and 5 min later a cold light source was placed on the skull at the sensorimotor cortex marker and was left in place for 20 min [44,45]. A subset of mice were subjected to sham stroke in which they underwent each step of the described procedure but with the light source turned off. Following induction of the stroke, the incision site was sutured, and the mice were monitored for recovery before being returned to the home cage.

### 4.3. Grip Strength Testing

A grip strength apparatus was built as described by Deacon [46]. We used varying lengths of chains of carabiners (5–10 carabiners) attached to a Scotch-Brite copper-coated scrubbing pad. Each carabiner (WODE Shop, Amazon Corp, Seattle, WA, USA) weighed 6.6 g and measured 47 mm in length. The Scotch-Brite pad weighed 3.8 ounces and measured 3.5 × 3.5 × 1 in. Each chain length was weighed and recorded. Three days following the stroke induction, each mouse was given successive trials with the varying carabiner chains, starting from the lightest to the heaviest, with five-minute rest periods between each trial. For each trial, the mouse was held by the tail and lowered until it was able to grip the scrubbing pad with the front paws. The mouse was then slowly lifted by the tail until the mouse held the apparatus above the table for a total of 3 s. The weight of the carabiner length which the mouse was unable to lift and hold for 3 s was recorded. If at any point the mouse gripped the apparatus with the hind paws or lifted for fewer than 3 s, the trial was reset.

### 4.4. Immunohistochemistry Staining

Three days following induction of stroke, mice were sacrificed and perfused with 1% PBS and 4% ParaFormaldehyde (PFA) solutions. The brain was removed and stored overnight in 4% PFA before being transferred to 30% sucrose solution. After adequate dehydration in 30% sucrose solution the brains were then further dried with a kimwipe and frozen in a matrix of OCT for slicing on the cryostat at 30 micromolar. The sliced tissues were left floating in 1% PBS solution and refrigerated until staining. The 30 micron slices were permeabilized with Triton x-100 in TBS, blocked for 60 min with 3% Donkey Serum, and received a primary stain of Iba-1 (1:500, Wako Chemicals USA, Richmond, VA, USA) for 24 h at 4C. After primary staining, the tissues were washed with TBS + Triton X-100 and stained with secondary antibody of Donkey Anti-Rabbit Alexafluor 488 (1:500, Abcam, Cambridge, MA, USA) at room temperature for 60 min. The tissues were washed once more with TBS and then mounted onto slides. Fluorogold + Dapi was used during mounting to stain cell bodies and reduce background fluorescence. Total fluorescence, microglial cell count, and microglial cell body size were analyzed using Image J software (National Institutes of Health, free download https://imagej.nih.gov/ij/download.html, accessed on 10 February 2021), and the analyzer was blinded to the experimental conditions. For microglial cell counts, total microglia within the field of the 20× microscope images from three slides for each animal were averaged, and this mean number for each animal in the experimental groups was averaged to determine mean total microglial number. For microglial cell body size, microglial cell body area from 10 microglia closest to the center of the 20× microscope images from three slides for each subject were determined and averaged using the Image J Process tool, and this mean number for each animal in the experimental groups was averaged to determine mean microglial cell body. Unit of measurements is pixel size predetermined by the Image J software.

### 4.5. Drugs

Beta-caryophyllene (Sigma Aldrich, St. Louis, MO, USA) and cannabidiol (Cayman Chemical, Ann Arbor, MI, USA) were each prepared in three concentrations (0.3, 1 and 3 mg/mL) in a vehicle mixture (1:1:18 by volume) of saline:Cremophor:ethanol and given as an intraperitoneal injection to create the respective treatment groups (3, 10 and 30 mg/kg). Vehicle was prepared with the same saline:Cremophor:ethanol (1:1:18) mixture and administered following the same experimental timing for each respective group. Rose Bengal (Sigma Aldrich) was dissolved in saline and administered IP at a volume of 0.1 mL of 10 mg/mL concentration.

## Data Availability

Not applicable.

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
