# Peer review of "Effects of Cannabidiol and Beta-Caryophyllene Alone or in Combination in a Mouse Model of Permanent Ischemia"

_ijms, 2021, doi:10.3390/ijms22062866_

Round 1

Reviewer 1 Report

This paper investigates the effects of Cannabidiol (CBD) and Beta-caryophyllene on permanent ischemia. Stroke is a major problem and new approaches to treat victims are needed. Generally, the manuscript is easy to read and comprehend and will be of interest to many readers given the recent legalization of marijuana in several states within the USA and approval of CBD as a drug. I advocate publication pending minor revision:

Line 56 –use italics for Cannabis sativa

Line 86 – take out "(CB2 receptor for BCP versus 5-HT1A or other receptor for CBD)." Could be poly-pharmacology as implied in text.

Line 121 – might be best to define what Iba-1 is and why it’s increase is a useful marker.

Line 138 – soma is used once but rest of the section uses cell-body. Please harmonize.

Figure 1 : Dunnett test used and Figure 2 uses Tukey. Please make consistent. 

Methods -- What is the source of Iba-1 ab?

Methods -- Imaging approach needs to be better described. How were the images quantified?

Methods -- How many animals per group? Authors state 80 mice were used. 

Discussion -- A prophylactic model was used for this study. This is not clearly stated and needs to be recognized. 

Author Response

Dear Reviewers

Thank you so much for the careful review of my submission. I appreciate the comments which have greatly improved the manuscript. All changes are highlighted in blue text in the resubmission. Here are my responses below:

Reviewer 1

Line 56 –use italics for Cannabis sativa

  • I have made this change

Line 86 – take out "(CB2 receptor for BCP versus 5-HT1A or other receptor for CBD)." Could be poly-pharmacology as implied in text.

  • This has been removed

Line 121 – might be best to define what Iba-1 is and why it’s increase is a useful marker.

  • I have added the following description “The Iba1 antibody is commonly used as a marker of microglia activation in staining and immunohistochemistry, given that ionized calcium binding adaptor molecule 1 (Iba1) is a microglia/macrophage-specific calcium-binding protein with actin-bundling activity that participates in membrane ruffling and phagocytosis in activated microglia.
  •  

Line 138 – soma is used once but rest of the section uses cell-body. Please harmonize.

  • I have changed all ‘soma’s to ‘cell body’s

Figure 1 : Dunnett test used and Figure 2 uses Tukey. Please make consistent. 

  • This was a Dunnetts and I have made the correction in the text

Methods -- What is the source of Iba-1 ab?

  • Wako, I have added this to the text

Methods -- Imaging approach needs to be better described. How were the images quantified?

  • I have added the following details ‘For microglial cell counts, total microglia within the field of the 20x microscope images from three slides for each animal were averaged, and this mean number for each animal in the experimental groups was averaged to determine mean total microglial number. For microglial cell body size, microglial cell body area from 10 microglia closest to the center of the 20x microscope images from three slides for each subject were determined and averaged using the Image J Process tool, and this mean number for each animal in the experimental groups was averaged to determine mean microglial cell body. Unit of measurements is pixel size predetermined by the Image J software.

Methods -- How many animals per group? Authors state 80 mice were used. 

  • We have added that n’s are 8/group

Discussion -- A prophylactic model was used for this study. This is not clearly stated and needs to be recognized. 

  • I have added this detail regarding the dosing regime in the introduction and discussion

Reviewer 2 Report

The present paper deals with the interesting issue and the results might be of interest but there are main points that need to be addressed.

Major Revision:

  1. Since both compounds alone have an effect at the highest dose of 30 mg/Kg, which is the rationale to use the fixed dose of 30 mg of BCP plus 3 or 10 or 30 of CBD?

  1. IBA-1 analysis and positivity should be better explained. Which is the difference between total fluorescence and microglial cell count?  Authors in Mat and Meth write: ImageJ software (NIH, Bethesda, MD, USA) was used to calculate the number of cells stained with IBA-1.

As a rule of thumb, Iba-1 stained cells are counted in the selected areas and results are usually expressed as the mean ± SEM of the number of reactive microglia/area (mm2). Moreover, IBA-1 positive cells can be classified in a resting or reactive microglial state according to their morphological aspects (e.g., resting cells with short branches or reactive microglia characterized by cell ameboid with a large soma and retracted and ticker processes).

The authors should explain the difference between total fluorescence and microglial cell count?  Which is the unit of measure of microglial cell count or body (#positive cells or pixel???)? Why the combination decreases the total fluorescence but not the microglial cell count and the microglial cell body size? Additionally, how they explain that CBD-30 alone increases the microglial cell body, while the combination of BCP: CBD led to a significant decrease?

  1. Finally, as expressed in the present form results are difficult to understand; e.g, In Microglia analysis: i) Total Fluorescence: Authors write: “Increased immunolabeling for IBA-1 was observed in the stroke animals compared to control” but in figure 2 it is not indicated sham control. Sham is no stroke in the graph? ii) Microglial cell count: Sham? No stroke in the graph? Moreover, no statistical values are given. There is no statistical difference in fig 2 (all panels) between no stroke and the other groups? Additionally, in the One-Way ANOVA for both CBD and BPC given alone the p values are lacking and the sign is < ; It is correct??? I guess wrong. Finally, authors should control letters in the graph of Fig 2 and their correspondence with the text (i.e., figures 2D and 3E in the text do not correspond with the graph, and p<0.05 are versus???).
  2. Discussion: “Alternately, protective effects of BCP at the doses tested here may be working through a micro-glia-mediated mechanism.” It is correct??
  3. Moreover, the advantages of using a combination of the two drugs versus single administration, if there are, should be stressed.  

Minor revision

Please correct misprinting such as “stoke” or statisitical”

Author Response

Dear Reviewers

Thank you so much for the careful review of my submission. I appreciate the comments which have greatly improved the manuscript. All changes are highlighted in blue text in the resubmission. Here are my responses below:

Reviewer 2

Major revisions

  1. Since both compounds alone have an effect at the highest dose of 30 mg/Kg, which is the rationale to use the fixed dose of 30 mg of BCP plus 3 or 10 or 30 of CBD?
  • As explained in the discussion, all doses, including the combinations, were selected a priori so that all animals could be run together. This is important as we see variability in stroke size across experiments and wanted to run all of the animals as closely together as possible. Doses were selected based on the literature with the scientific assumption that combinations selected would include single doses that were not effective on their own. While this ended up not being the case for the infarct size and grip strength, it was the case for microglial assessment. More detail on this was added to the discussion.

  1. IBA-1 analysis and positivity should be better explained. Which is the difference between total fluorescence and microglial cell count?  Authors in Mat and Meth write: ImageJ software (NIH, Bethesda, MD, USA) was used to calculate the number of cells stained with IBA-1.

  • We have added more detail in the methods, please see above response to reviewer 1. Measuring both total immunofluorescence and total count differentiate between activation of the microglia actually present versus an increase in the number of microglia/macrophages present at the site of injury. Additional measurement of cell body size was included as a proxy for changes in phenotype.

As a rule of thumb, Iba-1 stained cells are counted in the selected areas and results are usually expressed as the mean ± SEM of the number of reactive microglia/area (mm2).

  • This is what was done in the microglial cell number data

Moreover, IBA-1 positive cells can be classified in a resting or reactive microglial state according to their morphological aspects (e.g., resting cells with short branches or reactive microglia characterized by cell ameboid with a large soma and retracted and ticker processes).

  • We agree, our measurement of soma size was done to determine reactive microglia with large soma. We state in the discussion that this observation should be followed up with additional research to more fully characterize the phenotype of these microglia with larger soma size.

The authors should explain the difference between total fluorescence and microglial cell count?  

  • Measuring both total immunofluorescence and total count differentiate between activation of the microglia actually present versus an increase in the number of microglia/macrophages present at the site of injury.

Which is the unit of measure of microglial cell count or body (#positive cells or pixel???)?

  • Cell count unit is # of positive cells, cell body size is pixels determined by image J. This is explained more thoroughly in the methods section.

Why the combination decreases the total fluorescence but not the microglial cell count and the microglial cell body size?

  • The combination decreased total immunofluorescence and cell body size, but not the total number of microglia present, suggesting that treatment decreased a change in microglial phenotype but not a change in overall reactivity.

Additionally, how they explain that CBD-30 alone increases the microglial cell body, while the combination of BCP: CBD led to a significant decrease?

  • Again we believe this is due to CBD versus the combination leading to different phenotypic changes. It suggests on an IHC level, the combination does something very different to microglial than either drug alone (synergy). Beyond that, this is a mystery to us at this stage, but we see this exact same profile in other models in our laboratory, including chemotherapy induced neuropathic pain that we are in the process of writing up for publication. We believe this is discussed in detail in the discussion.  
  1. Finally, as expressed in the present form results are difficult to understand; e.g, In Microglia analysis: i) Total Fluorescence: Authors write: “Increased immunolabeling for IBA-1 was observed in the stroke animals compared to control” but in figure 2 it is not indicated sham control. Sham is no stroke in the graph? ii) Microglial cell count: Sham? No stroke in the graph? Moreover, no statistical values are given. There is no statistical difference in fig 2 (all panels) between no stroke and the other groups? Additionally, in the One-Way ANOVA for both CBD and BPC given alone the p values are lacking and the sign is < ; It is correct??? I guess wrong. Finally, authors should control letters in the graph of Fig 2 and their correspondence with the text (i.e., figures 2D and 3E in the text do not correspond with the graph, and p<0.05 are versus???).
  • We have changed the graphs to Sham and vehicle to match the text and have clarified that sham are no stroke and vehicle are stroked and treated with vehicle. We have also clarified that for statistics, we first ran t-tests to determine differences between sham and stroke/vehicle, and one-way ANOVA to analyze treatment effects in stroked mice (veh versus cannabinoid treatment). Figure correspondences in the text have also been corrected.
  1. Discussion: “Alternately, protective effects of BCP at the doses tested here may be working through a micro-glia-mediated mechanism.” It is correct??
  • No, thank you for catching this error! We have changed this to microglia-independent
  1. Moreover, the advantages of using a combination of the two drugs versus single administration, if there are, should be stressed.  
  • We have added this to the introduction

Minor revision

Please correct misprinting such as “stoke” or statisitical”

  • These have been corrected

Round 2

Reviewer 2 Report

In this revised form the paper is acceptable for publication